# Insight into the Role of Psychological Factors in Oral Mucosa Diseases

**DOI:** 10.3390/ijms23094760

**Published:** 2022-04-26

**Authors:** Yuexin Guo, Boya Wang, Han Gao, Chengwei He, Rongxuan Hua, Lei Gao, Yixuan Du, Jingdong Xu

**Affiliations:** 1Department of Oral Medicine, Basic Medical College, Capital Medical University, Beijing 100069, China; gyxin2014@163.com (Y.G.); duyixuan0312@163.com (Y.D.); 2Department of Clinical Medicine, Peking University Health Science Center, Beijing 100081, China; 1810301208@pku.edu.cn; 3Department of Physiology and Pathophysiology, Basic Medical College, Capital Medical University, Beijing 100069, China; gaohan703851@163.com (H.G.); hcw_1043@163.com (C.H.); 4Department of Clinical Medicine, Basic Medical College, Capital Medical University, Beijing 100069, China; andrewhdd@126.com; 5Department of Bioinformatics, College of Bioengineering, Capital Medical University, Beijing 100069, China; bmi5@ccmu.edu.cn

**Keywords:** psychology, oral mucosa, disease, interactions, pathology

## Abstract

With the development of psychology and medicine, more and more diseases have found their psychological origins and associations, especially ulceration and other mucosal injuries, within the digestive system. However, the association of psychological factors with lesions of the oral mucosa, including oral squamous cell carcinoma (OSCC), burning mouth syndrome (BMS), and recurrent aphthous stomatitis (RAS), have not been fully characterized. In this review, after introducing the association between psychological and nervous factors and diseases, we provide detailed descriptions of the psychology and nerve fibers involved in the pathology of OSCC, BMS, and RAS, pointing out the underlying mechanisms and suggesting the clinical indications.

## 1. Introduction

The oral cavity is the entrance of our digestive system, and accounts for the generation of our taste [1]. Despite its small size, the tissue structure and nerve innervations of the oral cavity are complex and form close interactions with the brain, as well as the whole body [2]. The food we eat can exert an impact on our emotions, and lesions in the oral mucosa can seriously affect our daily life. Although several diseases of the oral mucosa are common in a wide number of people of different ages and areas, their detailed pathologies are largely unknown [3]. This might limit the process of drug development, as well as the treatment effects. Based on our current understanding, several factors such as dietary habits and food stimulation might account for disease genesis and pathology. Other factors including immune reactions and psychological conditions have also been raised, although knowledge of them is largely limited. In the meantime, there are distinctions between clinicians and patients considering the definition of treatment success. Most patients consider it as the relief of symptoms and the recovery of their functions, while clinicians may put more emphasis on tissue repair and the radical cure of disease course with the assistance of histological and imaging technology [4]. This difference may increase the difficulty in permanent cures and the long-term maintenance of treatment which has been a growing quandary for many clinicians.

Psychological factors are significant mediators of several diseases’ pathologies (as shown in Table 1, Table 2 and Table 3 for their relations, respectively) and form close interactions with multiple tissues and systems. Although they are most recognized by psychologists and utilized by means of mental consultation, their biological foundations are also recognized by biologists and doctors, promoting their use in drug development and clinical treatment. Known as the bio–psycho–social medical model first put forward by George Engel in 1977 [5], psychology has received more and more attention in the genesis, progression, and treatment of the medical process. As a complex system regulated by various areas in the human brain, the psychology and mental health of people are in direct contact with the nervous system. Considering the broad functions of neurotransmitters (NT) and the wide distribution of their receptors, not only could they form close interactions with body immunity by the activation or inhibition of immune pathways, but also regulate the functions of normal tissues with the initiation of physiological reactions [6]. In the meantime, they are also essential in the regulation of disease pathology, as an increasing number of diseases have been found to be associated with psychological origins or manifestations, such as gastritis [7] and breast cancer [8]. However, despite the broad and deep investigations of interactions between the nervous system and other organs, those of the oral mucosa are relatively scarce with no comprehensive analysis.

**Table 1 ijms-23-04760-t001:** Oral symptoms involved in psychological problems.

Psychology Problems	Oral Symptoms	Refs.
Olanzapine-induced anticholinergic toxicity	Dry oral mucosa	[9]
Alzheimer’s disease (AD)	Tau protein in oral mucosa	[10]
Bulimia and anorexia nervosa	Abrasion of teeth enamel	[11]
Epidermoid cysts of the central nervous system	Similar to symptoms in oral mucosa	[12]
Neurocutaneous syndromes (and other diseases associated with DNA repair)	Teeth and oral mucosa lesions	[13]
Structural changes in innervations in oral cavity	Loss of sense	[14]

**Table 2 ijms-23-04760-t002:** Oral diseases with psychological abnormalities.

Oral Diseases	Psychological Abnormalities	Oral Manifestations	Refs.
Oral squamous cell carcinoma (OSCC)	Higher α1 adrenergic receptors	Oral ulcers and lumps, pain feeling	[15]
Primary Sjögren’s syndrome	Depression and anxiety	More frequent oral lesions; negative impact on life quality	[16]
Burning mouth syndrome (BMS)	Structural and functional deficits within the nervous system	Burning feeling without an obvious cause	[17]
Herpes simplex encephalitis	Fatal disease of the central nervous system	Painful blisters or open sores (ulcers)	[18]
Oral mucosa cancers	Antitumor drugs improve psychological symptoms	Oral pumps and lesions	[19]
Primary burning mouth syndrome	PNS involvement	Burning feeling in mouth	[20]
Recurrent aphthous stomatitis	Hypofunction of the sympathetic nervous system	Painful round shallow ulcers	[21]
Inflammatory stimulation of the oral mucosa	Activation of microglial cells	Oral mucosal inflammation	[22]
Poliovirus	Affects the anterior horn motor neurons of the spinal cord causing paralysis		[23]
Xerostomia	Autonomic nervous system imbalance	Dryness of the oral mucosa	[24]
Burning mouth syndrome	Decreased or modified steroid synthesis	Burning feeling in mouth	[25]
Lingual conical papillae	Alterations of different kinds of neurons	Bumps and rough tongue	[26]
Lichen planus and lichenoid reactions	Loss of PNS fibers	Asymptomatic white reticular striae to painful erythema and erosions	[27]
Heat stimulation	Large primary neurons responding to high-threshold noxious heat are abundant in the tooth pulp	Altered pain	[28]
Oral mucosa continuous remodeling	Sensory nervous apparatus involvement	Leaky epithelial barrier, a fibrotic lamina propria, the release of inflammatory mediators, and the recruitment of immune infiltrate	[29]
Oral dysesthesia (OD)	Soft tissue grafts	Merkel cells and permanent dysesthesia in the oral mucosa	[30]

**Table 3 ijms-23-04760-t003:** Diseases with both the psychological problems and oral manifestations.

Diseases	Psychological Problems	Oral Manifestations	Refs.
Congenital herpes simplex	CNS infection	Oral infection	[31]
Enterovirus A71 (EV-A71)	Viral antigens/RNA in the CNS	Viral antigens/RNA in the squamous epithelia of the oral cavity	[32]
Verrucous lesions	Malocclusions in CNS	Verrucous lesions affect the oral mucosa (rare)	[33]
Cryptococcosis	Presentations in the CNS	Excision of nodules in the oral mucosa assists recovery	[34]
Paracoccidioidomycosis (PCM)	Involvement of CNS	Common infected symptoms	[35]
Herpes simplex virus type 1 (HSV-1)	Clinical diseases in CNS	Vesicular lesions of the oral mucosa	[36]
Cowden syndrome	Similar CNS symptoms with that in the oral mucosa	Multiple hamartomatous neoplasms of the oral mucosa	[37]
Lipoid proteinosis (LP)	Involvement of CNS	Yellow-white plaques on oral mucosa	[38]
Enterovirus 71 (EV71) (hand-foot-and-mouth disease (HFMD))	brainstem encephalitis	Vesicular lesions on oral mucosa	[39]
Wilson disease (WD)	multi-organ manifestations involve the nervous system	Repeated oral candidiasis	[40]
HSV-1	Transmission in the CNS	Infection in the oral mucosa	[41]
Tuberous sclerosis	Hamartoma formation in the nervous system	Hyperpigmented and hypopigmented macules affecting the oral mucosa	[42]
Bacillary angiomatosis	CNS	BA lesions in the oral mucosa	[43]
Cowden disease (CD)	CNS manifestations	Normal oral involvement	[44]
Dettol liquid	CNS symptoms	Oral involvement	[45]
Sweet’s diseases			[46]
HSV-1 infection	Vagus nerve transmission	Oral manifestations	[47]
Adamantiades–Behçet disease	Lesions of ulcerating systemic vasculitis in the CNS	Oral manifestations	[48]

In this review, after briefly introducing the broad effect of psychological molecules in the whole body, we put our emphasis on the roles of psychological factors in the generation and progression of oral mucosa injuries, especially those of OSCC, BMS, and RAS. Besides this, we highlight the similarities in their roles in pathogenesis, summarizing the related factors and providing the general interactive patterns. Furthermore, we analyze the current understanding of the interactions of oral diseases and psychological factors and put forward the directions for future research. These would be of great use considering the broad roles of psychological factors in the whole body, and would provide novel and practical suggestions for clinical treatment.

It has been recognized that receptors for adrenalin and its derivatives are widely distributed in a variety of tissues and organs, and employ multiple functions all over the body [49]. Their basic roles lie in the activation of neurons and the initiation of neuro-excitement. However, they are also responsible for muscle contraction and gland secretion [50,51]. Further investigations have also shown their roles in immune cell activation and proliferation, and the transmission of typical feelings [52]. Common neurotransmitters are responsible for these functions, including adrenalin and several amino acids. These regulations form a close regulatory network in our body which is only partly understood due to the limitations of our current knowledge and techniques.

## 2. Roles of Psychological Factors in Oral Mucosal Diseases

### 2.1. Psychological Factors in the Progress of OSCC

OSCC is one of the most malignant tumors around the world, with an overall 5-year survival rate of approximate 50–60% [53]. Common symptoms of OSCC include vomiting and swallowing abnormalities, and seriously affect patients’ normal lives [54]. Although smoking has been confirmed as the most dangerous factor associated with OSCC [55], multiple factors have been corroborated in association with OSCC pathology and a significantly higher level of distress thermometer (DT) score has been found in OSCC patients [56]. Several biomarkers could be utilized for the evaluation of OSCC progression and malignancy, some of which are morphological characteristics of cells such as tumor-infiltrating lymphocytes (TILs), as well as immune checkpoints such as PD-L1, FKBP51, B7-H4, B7-H6, ALHD1, IDO1, and B7-H3 [57]. Other studies have shown the potential of osteopontin and fractal dimension in predicting diagnoses and choosing the proper treatment in OSCC [58]. Further analysis has found an increase in the expression of α1 adrenergic receptors (α1-ARs) in serum and saliva lesions in oral mucosa [15], mainly accounting for carcinoma cell proliferation and translocation. Concomitant with this, exposure to nicotine has been found to induce OSCC proliferation [59], which has also been corroborated in other carcinoma models [60]. In the meantime, various studies have also shown the stress-related upregulation of α1-ARs in both human beings and animal models [61,62]. Multiple pathways could be activated via increasing the expression of α1-ARs such as the mitogen-activated protein kinase (MAPK) pathway, cAMP metabolism, and phospholipase D (PLD) and A2 in different cells, and Nishioka et al. confirmed the activation of the EFGR/ERK/AKT pathway in the course of the disease [59]. However, further analysis is also required concerning the potential roles of other nicotine-related pathways, and their long-term detection and examination are also needed.

Another significant difference involving the nervous system is the increase in salivary cortisol level, indicating the disturbance of the hypothalamus pituitary axis [63]. This is also related to the norepinephrine level, as Lee et al. found a higher cortisol level to be induced in children with more variability after stress and following the secretion of norepinephrine [64]. In the meantime, both α1 adrenergic receptors and cortisol account for the significant anti-inflammation roles of NSAIDS [65], and the induction of hydrocortisone results in alterations of the cardiac adrenergic receptor density which are associated with the functional outcome in rats [66]. Further analysis has confirmed a similar effect of hydrocortisone in increasing the sensitivity to α1 adrenergic receptors in humans after hemorrhagic shock [67]. However, an induction of 0.1 mM hydrocortisone prohibited the decrease in 3H-clonidine binding sites with no impact on 3H-yohimbine or the 3H-prazosin binding sites in rat organ culture [68], and Prazosin pretreatment did not affect the basal or peak plasma cortisol level during hypoglycemic stress in human beings [69]. These studies show the complex interactions between α1 adrenergic receptors and the cortisol level which could be affected in dose- or tissue-dependent ways. Further investigations would be of great help in uncovering the detailed interactions between them and their potential use in clinical treatment.

Intensive investigations into neurotransmitters and other neurokinins have found that neurokinin B (NKB) can stimulate the proliferation of OSCC, probably via the pAkt/pmTOR signaling pathway [70]. Additionally, this is concomitant with the different expression of the neurokinin 3 receptor (NK-3R) in both the central and peripheral nervous systems in OSCC [71]. In the meantime, Obata et al. found a higher expression of tachykinin 3 (TAC3) in the invasion front of oral squamous cell carcinoma in bone matrix; TAC3 (Tachykinin-3) is probably released by the peripheral sensory nerves and contributes to tumor progression [71]. Gamma-amino butyric acid (GABA), a negative neurotransmitter in the nervous system, is also found to promote the proliferation of OSCC via the activation of the p38 MAPK and inhibition of the JNK/MAPK signaling pathways [72]. Concomitant with this, glutamate acid decarboxylase 1 is reported to promote the metastasis of human oral cancer by β-catenin translocation and MMP7 activation [73]. PrPC (cellular prion protein), well known for its roles in neurodegenerative diseases, has also been corroborated for the ability to resist tumor necrosis factor α (TNFα) apoptosis in OSCC, colonic, and the renal adenocarcinoma ACHN [74]. Say et al. further showed this enhancement in the expression of PrPC resulted from stable snRNA knockdown and could lead to the inhibition of glycosylation [75]. Semaphorin7A, a chemotactic factor in neurogenesis as well as a significant immunomodulator, has been found to promote tumoral growth and metastasis in human oral cancer by activating the G1 cell cycle and matrix metalloproteases [76]. Natriuretic peptide systems, which are important regulators of the nervous system and in controlling the secretion of saliva, also show abnormalities in OSCC probably related to the disturbance of the water-salt imbalance [77]. Taken as a whole, these molecules may work together and form a series of cascade reactions that contribute to perineural invasion (PNI), which might facilitate the formation of a tumor microenvironment (TME) and tumor metastasis [78,79].

With the development of gene research, more investigations have reported the significant roles of ncRNA in regulating the expression of genes encoding proteins associated with nerve-related tumor progression. LncRNA could induce the activation of the NF-κB/STAT3 pathway and the secretion of proinflammatory cytokines [80], and miRNAs are suggested as markers in OSCC, as the overexpression of miR-181a and miR-181b may increase lymph-node metastasis and vascular invasion and is associated with a poor prognosis in OSCC patients, while the downregulation of miR-125b has been found in OSCC cell lines [81]. Despite the lack of genetic level studies nowadays, more investigations would be of great help in understanding the mechanistic interactions with the development of molecular biology.

### 2.2. Burning Mouth Syndrome (BMS)

Known as a common disease affecting almost every 1 in 10,000 people, BMS often occurs with no typical symptoms. However, many patients show a loss of/reduced taste perceptions during the disease course which is often considered to be in close connection with psychological factors, such as depression and anxiety [82]. Simultaneously, the severity of its syndrome is also closely correlated with patients’ mental condition [83]. Meta-analysis also shows lower cold detection and pain thresholds, as well as higher warm detection and pain thresholds at the tongue and lip of BMS patients compared with healthy participants. However, no significant differences in mechanical detection and pain thresholds were found [84]. Superficial biopsies of the lateral aspect of the anterior two-thirds of the tongue showed a significantly lower density of epithelial nerve fibers with diffusing morphological changes that reflect axonal degeneration [85]. In the meantime, the expression of the transient receptor potential vanilloid 1 (TRPV1) and P2 × 3 receptor are upregulated and account for the generation of pain feeling [86]. Personality analysis also confirmed a tendency of BMS patients to be more introversive and unstable compared to the healthy groups [87]. In addition, their plasma norepinephrine level was also higher which might account for the fiercer reactions after stimuli. BMS is also known as a disease affecting various parts of the body and is witnessed as a manifestation of diabetes. However, despite the small distinction of several prognostic indexes between the placebo spray (PS) and artificial saliva (AS), no significant difference was found according to statistical methods (the indexes include salivary flow rate, antioxidant capacity of saliva, and ultrasound variables) [88].

Simultaneously, studies focusing on abnormalities of the central nervous system have also shown an altered structural connectivity of pain-related brain networks using graph analyses of probabilistic tractography [89]. Further research has shown that six out of eight regions of the gray were affected (anterior and posterior cingulate gyrus, lobules of the cerebellum, insula/frontal operculum, inferior temporal area, primary motor cortex, dorsolateral pre-frontal cortex (DLPFC)). In the anterior cingulate gyrus, the lobules of the cerebellum, the inferior temporal lobe, and the DLPFC, pain intensity is correlated with gray matter concentration [90]. Concomitant with the reduced volume of gray matter, cerebral blood flow (CBF) also decreased which is related to the pain severity [91]. Apart from the alterations mentioned above, Jääskeläinen et al. found a hypofunction of dopaminergic neurons in the basal ganglia in a subgroup of people. This suggests the complexity of this regulatory network and the need for more detailed examination before clinical practice [92]. Further investigations have shown a more comprehensive regulation of pain feeling via the dopaminergic nervous system, as dopamine can act on striatal dopamine D2/D3 receptors and serotonin on cortical 5-HT1A and 5-HT2A receptors and affect top-down pain regulation in humans [93].

Besides the manifestations within the central nervous system, research has also revealed that alterations in the oral mucosa are in accordance with several changes in the central nervous system. The loss of peripheral nervous fibers is concomitant with the lower threshold of pain detection which has suggested the roles of neuroinflammation in oral nerve defects [94] and is in line with the positive effect of benzodiazepines in BMS patients. Based on the analysis above, although the detailed and comprehensive mechanisms underlying BMS have not been fully characterized, the roles of several typical molecules have been confirmed. This has laid the foundation for drug development for conditions such as idiopathic glossodynia which could exert an altered excitability in the trigeminal nociceptive pathway in the peripheral and/or central nervous systems, and enhance GABA concentration while decreasing glutamate functions at postsynaptic sites. Clinical uses of several such drugs, for example, topiramate, have been evaluated and received good feedback in some patients [95]. Additionally, neurophysiological evaluation as well as psychological tests for BMS found that sufferers were characterized by a mild sensory and autonomic small fiber neuropathy with concomitant central disorders [96]. A frequency analysis of the heart rate variability (HRV) revealed autonomic instability and the tracking of these changes corrected with stellate ganglion near-infrared irradiation (SGR) also provided a novel evaluation for follow-up survey, as well as a therapeutic measurement for BMS patients [97]. Simultaneously, based on the findings that chronic anxiety could result in a dysregulation of the adrenal cortex’s production of steroids, the decreased production of some major precursors for neuroactive steroid synthesis and the resulting brisk alteration of neuroactive steroid production might contribute to the neurodegenerative alterations of small nerves fibers of the oral mucosa and/or brain areas involved in oral somatic sensations [25]. These neuropathic changes could probably become irreversible with disease progression and result in burning pain, dysgeusia, and xerostomia associated with stomatodynia from thin nerve fibers. Considering the possible complications of BMS, Varvet et al. also suggested the potential roles of the dopaminergic nervous system in a case of Lewy body [98]. Another research study has also reported the defects of dopaminergic neuron in the central nervous system in BMS. This offers the potential for interventions targeting pathophysiological mechanisms and related molecules [92]. Moreover, researchers also made the comparation between BMS and Parkinson’s disease due to the similar intensity of autonomic nervous system dysfunction. A significant impairment of both the sympathetic and parasympathetic nervous systems was found with the maintenance of sympathetic/parasympathetic balance in BMS patients [99]. This indicates the probable existence of common regulatory mechanisms between them which are waiting for further elucidation. Another case of a diabetic patient revealed the roles of trigeminal nociceptive pathways in BMS pathology [100]. As an essential regulator of maxillofacial sensation, trigeminal sensory fibers have received much attention and are worth the research considering the wide innervated regions and broad functions. Studies concerning the roles of oral immunity in BMS have also shown the wide involvement of the immunoendocrine system which could mainly and specifically account for depression in BMS. Although the activation of the hypothalamic–pituitary–adrenal axis and the sympathetic nervous system are predominantly due to psychological stress and are not specific to BMS, the immunoendocrine mechanism in BMS is worth attention regarding its broad roles in a variety of diseases [101].

Tricyclic antidepressants (nortriptyline and amitriptyline), serotonin-noradrenaline reuptake inhibitors (SNRIs) (duloxetine and milnacipran), and antiepileptic drugs including potential-dependent calcium channel α2δ subunit ligands (gabapentin and pregabalin) which are regarded as the first-choice drugs for neuropathic pain under current standards. However, their effects are not the same for different patients in various pathological conditions. Additionally, pregabalin might be a novel option for BMS patients who are resistant or not responsive to SNRIs due to its different activating mechanism [102]. Despite the use above, the general effectiveness and safety of psychological treatment in BMS have not been fully confirmed, as Eccleston et al. found no significant variation between the control group and those treated with psychological interventions [103]. Other research works focusing on neuroprotective steroids are still in the trial period with no unified conclusions [104].

Apart from direct disorders of the nervous system, other oral manifestations may also be associated with neuro disorders and provide potential clinical indications such as the change in microbial amounts and components. This is concomitant with findings that BMS is not limited to oral diseases, but has close interactions with general body health [105]. According to the human oral microbiome database (HOMD) [106], an approximation of 700 prokaryote species are localized in the oral cavity [106]. These microbes are closely related with the body health condition of human beings and show relative alterations in times of disease [107]. This alteration might be achieved by microbial metabolic products which could be utilized as neurotransmitters or antibodies, and initiate a serious of host actions [107]. These interactions might also account for the different pathologies and outcomes of comorbidities, as the amount of bacteria in tongue mucosa is significantly reduced in benign migratory glossitis compared with atrophic glossitis and BMS. The change in mucosal bacteria is associated with morphological alterations, making the oral environment more acceptable for H. *pylori* (HP) colonization and facilitating oro-oral transmission [108]. In the meantime, HP could also alter the pathogenesis of BMS, as a majority of BMS patients recovered after HP elimination [109]. The potential roles of HP in the generation and progression of oral pain have also suggested its probable relation with BMS [110].

### 2.3. Recurrent Aphthous Stomatitis (RAS)

RAS, characterized by rounded shallow painful ulcers with a yellowish gray pseudomembranous center and a well-defined erythematous rim, is the most common ulcerative disease of the oral mucosa in clinical practice [111]. Similar to other ulcers including ulcerative colitis (UC) and gastric ulcers, RAS occurs in approximately 2% to 10% of Caucasian populations and seriously affects their life quality and work efficiency [112]. The relationship between RAS and psychology has also received much investigation and shown its roles in RAS pathology [113]. Intriguingly, despite the close relationship of depression and salivary cortisol level [114], no significant difference was found between the RAS patients and the control group [115,116]. However, when the *p* value was set as 0.01 rather than 0.05 to indicate the difference, a distinction was found in an experiment with a similar design [117]. Other studies have also reported a higher level of cortisol in both the saliva and serum of RAS patients [118]. Catecholamine concentration shows a similar trend [119]. This is similar to that of OSCC and BMS, and suggests a similar mechanism underlying them. Further investigations have shown the hypofunction of the sympathetic nervous system in RAS patients [21] with a significantly higher level of salivary α-amylase enzyme (sAAE) in RAS patients [120]. As a common biomarker in the nervous system, sAAE is associated with stress, anxiety, and defects of the nervous system [121]. In the meantime, sAAE is also associated with alterations of microbial components and diversity, as Zulfiqar et al. found that *F. nucleatum* cells inhibited the enzymatic activity of salivary α-amylase in a dose-dependent manner [122]. sAAE is even utilized as a bio-marker for RAS patients, especially when evaluating their stress level [121]. Stress in RAS patients might also account for the alteration of the promoter region of the serotonin transporter (5-HTT) and result in decreased transcriptive activities for serotonin expression [118,119,123].

In addition to their roles in RAS pathology, several psychological factors are also important in the prognosis of RAS [124]. However, related research works are scarce and the detailed mechanisms underlying them still require further investigation.

All these findings suggest the essential roles of bacteria in the pathogenesis of oral mucosal diseases. Here, we discuss three typical and serious roles as representative and apart from these, other oral symptoms involved in psychological problems are summarized in Table 1. From the analysis above, several similarities for psychological factors in the pathology of oral diseases could be extracted. To begin with, the pathologies of many oral diseases are associated with psychological factors, and many oral manifestations are accompanied by nerve dysfunctions and/or degeneration alterations. This lays the foundation for their interactions, and is found to be relative to changes in central nervous system and/or the peripheral nervous system. Moreover, changes in the nervous system are accompanied by alterations of the oral bacteria. This increases the relation between oral diseases and overall body health, and might also account for the comorbidities of certain diseases. In addition, psychological factors, especially small molecules such as neurotransmitters, are of potential use in clinical treatment. These lay the basis for psychological interventions by means of drug therapy and mental consultation. However, further investigation is required before effective use due to the complex regulatory mechanisms and potential dose-dependent regulatory pathways.

## 3. Psychological Factors Affected by Oral Diseases and Dietary Habits

Studies have corroborated that the food we eat every day is closely associated with the feelings we have. Simultaneously, they might also become the origin of several diseases, as some oral diseases are accompanied with psychological abnormalities as shown in Table 2. Studies have corroborated that these impacts are achieved via several pathways similar to the ones accounting for several pathological processes [125]. Additionally, the key regulatory molecules also show much resemblance. The well-known TRPV channel conducting the spicy sense depends on the activation of CB1 and CB2 receptors on the neuronal synapse in the brain. OTOP1 receptors are responsible for the sour taste, and SEMA3A and SEMA7A are shown to be significant transmitters in bitter and sweet taste neurons for taste-receptor cells (TRCs) [126]. The activation of these key factors in oral diseases might serve as the initiators of alterations in the nervous system and further result in psychological changes. Some typical preferences are characteristic of certain diseases; for example, most patients of ulcerative colitis often feel stressed and fond of spicy food [127], although detailed relations remain unknown. Other common preferences for food taste, including those for the most common sweet and sour food, have been summarized [128] and even utilized for treatment [129]. The effects of these foods varies a lot among people in both age and sex, as studies found a higher preference for sweet for women, which is also related to typical physical status including menstrual period and estrogen level [130]. Adolescents show a similar tendency for sweets with a stronger desire especially in times of being lured by the outer environment [131]. Heavy smokers are taught to have some sweets in times of craving hits, and those with diabetes are found to have a decrease in food taste and sweet perception [132]. All these phenomena are of potential use in clinical treatment and would bring bright prospects for patients, considering the increase in life quality and treatment outcomes.

The interactions among psychological factors are also worth attention, as the occurrence is often a mixture of several bad feelings which could work together and strengthen the neuro feedback. This exacerbation might aggravate the pathology of oral mucosa and result in a bad prognosis. These reactions could be mediated by both local tissues and signals from the central nervous system, and could exert an impact on multiple systems. Moreover, they are also likely to be of great benefit for patients concerning their replacement of drugs with everyday food.

## 4. Information Is Analyzed in the Brain and Sent Back to Oral Mucosa

Despite the large effort in understanding how information is analyzed and comprehended within the brain, clear and convictive mechanisms remain unknown. However, the existence of neurotransmitters in oral tissue and receptors on oral glands and epithelial cells have been confirmed, and several typical feedback mechanisms between oral tissue and the nervous system have been elucidated. The sensitivity and amount of these receptors are related to the extent of the stress and psychological conditions of individuals. This regulation is mediated via both the local tissues and interactions within the brain. Significant roles of endocannabinoids (ECS) have been reported in the mediation of this information transportation, with PPAR-α and TRPV being their co-workers [133]. Current knowledge of these interactions is mainly limited in different neurotransmitters in different part of the brain and types of neurons, respectively, while those elaborate innervations are largely unveiled. More research is worthwhile concerning its potential uses in understanding the pathology of and providing treatment for diseases of both the oral mucosa and psychological well-being. In fact, many diseases come with both the psychological problems and oral manifestations (as shown in Table 3), and are worth more research considering the potential treatments it might bring forward.

## 5. Conclusions

Psychological factors are of great importance in clinical treatment under our current knowledge. They are also of great use in the treatment of oral mucosal diseases and the development of related drugs. However, the current understanding of their detailed mechanisms is largely limited to the level of key molecules. In this review, after briefly introducing the general interactions between psychological and nervous system factors, we have provided detailed information about the three most common diseases in oral mucosa. We have also put forward relative disease pathologies based on current knowledge and analyzed the similarities among them. Besides this, we have summarized the type of oral lesions associated with psychological and nervous system problems, pointing out the potential clinical prospects for clinical diagnosis and treatment. The knowledge of these key molecules is also important for drug development, and offers great help in relieving patients’ symptoms and increasing their life quality. Despite our efforts to generalize the common roles of psychological factors in the progression of oral mucosal diseases, there might also be some limitations. For example, whether similar drugs could be utilized in different disease models and the specific dosages of these is waiting to be evaluated. Moreover, psychological treatment is mainly limited to mental consultations, while chemical drugs are relatively scarce. This may be due to the lack of molecular-targeted investigations on drug development of psychological diseases. However, with the development of psychology and medical technology, more genetic methods would be of great help in understanding the underlying mechanisms in disease pathologies and offering psychological methods for the treatment of oral mucosal diseases.

## Data Availability

Not applicable.

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
