# Peer review of "Insight into the Role of Psychological Factors in Oral Mucosa Diseases"

_ijms, 2022, doi:10.3390/ijms23094760_

Round 1
Reviewer 1 Report
I find the topic interesting and bringing some novelty although the construction of the manuscript could be improved, sometimes I do not follow what the Authors wanted to present.
Eg. I don't understand table 2. In the first column diseases are mentioned but in the second it says there are 'psychological abnormalities' whereas there are different symptoms mentioned and some of them don't fit into this headline at all. Please explain the purpose of this table.
Moreover, suddenly Box 1 appears in the line no 80, where there's no box?
There is not section on materials and methods. The Authors should describe how they performed their review and what the search strategy was. Ideally, they should include a diagram.
The references include over 100 articles, which is an advantage, but they are prepared entirely not according to the Journal's guidelines, even including the font.
English has to be improved, although it's readable, there are many mistakes and sometimes it's difficult to follow.
Author Response
Responses to reviewer 1
I find the topic interesting and bringing some novelty although the construction of the manuscript could be improved, sometimes I do not follow what the Authors wanted to present.
Eg. I don't understand table 2. In the first column diseases are mentioned but in the second it says there are 'psychological abnormalities' whereas there are different symptoms mentioned and some of them don't fit into this headline at all. Please explain the purpose of this table.
Response: Thanks for your generous comment and pointing our limitations. We intend to show the name of diseases related to both the oral disorder and psychological manifestation in the first column, and in the second one we intend to show typical psychological disorder involved. We have added several details and obliterate the irrelevant content. (changes are shown in bright blue)
Moreover, suddenly Box 1 appears in the line no 80, where there's no box?
-Response: Thank you for your kind remind. The following paragraph is the content of box 1 and we have increased the font in order to have a better illustration.
There is not section on materials and methods. The Authors should describe how they performed their review and what the search strategy was. Ideally, they should include a diagram.
- Response: Thank you for your kind advice. We did not include detailed search strategy for it was not designed to be a systematic review or meta-analysis from the beginning. But we have added the mind map to show the outline of our review.
The references include over 100 articles, which is an advantage, but they are prepared entirely not according to the Journal's guidelines, even including the font.
Response: Thank you for your kind remind. We have changed the reference according to the Journal's guidelines.
English has to be improved, although it's readable, there are many mistakes and sometimes it's difficult to follow.
Response: Thank you for your kind suggestion. We have polished our language and corrected the mistakes in order to make the paper more understandable(as shown in bright blue).
Reviewer 2 Report
The article entitled “Roles of psychological factors in oral mucosa diseases” aimed to investigate the association between psychological and nervous factors and oral diseases, including the oral squamous cell carcinoma (OSCC), the burning mouth syndrome (BMS) and recurrent aphthous stomatitis (RAS). The authors suggested clinical indications related to these conditions. The paper is in line with journal’s aim, moreover, Authors have well revised several issues; however, I ask authors to add some key concepts.
- In the title it is necessary to specify the type of study
- In the section entitled “Psychological factors in the progress of OSCC” the authors should highlight the presence of the pathogenetic processes of the OSCC of other pathognomic biomarkers of the advanced degree of tumor malignancy (please see and discuss PMID: 30043590).
- In the section entitled “Burning mouth syndrome (BMS)” the authors should stress the oral health-related quality of life related to symptoms such as dry mouth (xerostomia) which can influence the psycho-social sphere of the affected patient (please see and discuss PMID: 32664567
- The limitations of the study must be added in the paper
- Research suggests how patients and clinicians define the concept of success differently.
- Conclusions cannot be reduced to a sentence: you must improve them highlighting the limits and the future insights pointed out from this article.
According to this Reviewer’s consideration, novelty and quality of the paper, publication of the present manuscript is recommended after minor revision.
Author Response
Responses to reviewer 2
The article entitled “Roles of psychological factors in oral mucosa diseases” aimed to investigate the association between psychological and nervous factors and oral diseases, including the oral squamous cell carcinoma (OSCC), the burning mouth syndrome (BMS) and recurrent aphthous stomatitis (RAS). The authors suggested clinical indications related to these conditions. The paper is in line with journal’s aim, moreover, Authors have well revised several issues; however, I ask authors to add some key concepts.
In the title it is necessary to specify the type of study
Response: Thank you for your kind suggestion. We have changed the title and added “a review” for specification.
In the section entitled “Psychological factors in the progress of OSCC” the authors should highlight the presence of the pathogenetic processes of the OSCC of other pathognomic biomarkers of the advanced degree of tumor malignancy (please see and discuss PMID: 30043590).
Response: Thank you for your kind suggestion. We have added relevant part in bright blue in 108-123.
In the section entitled “Burning mouth syndrome (BMS)” the authors should stress the oral health-related quality of life related to symptoms such as dry mouth (xerostomia) which can influence the psycho-social sphere of the affected patient (please see and discuss PMID: 32664567
Response: Thank you for your kind suggestion. We have added relevant part in bright blue in 191-195.
The limitations of the study must be added in the paper
Response: Thank you for your kind suggestion. We have added relevant part in bright blue in 386-392.
Research suggests how patients and clinicians define the concept of success differently.
Response: Thank you for your kind suggestion and we have added that part as shown in bright blue in line 46-52.
Conclusions cannot be reduced to a sentence: you must improve them highlighting the limits and the future insights pointed out from this article.
Response: Thank you for your kind suggestion. We have added that part in bright blue in line 386-392.
Round 2
Reviewer 1 Report
The Authors improved the manuscript, although I still have some remarks.
I still don't understand the purpose of 'box 1' - couldn't it be another numbered paragraph?
I still don't see the materials and methods section. The Authors say that they added such part, but I can't really find this in the text. Please add a distinct section and describe how you searched articles, what MeSH you used and what languages were taken into account.
References are still not prepared according to Journal's guidelines. In the text they are also superscripted, not in the square brackets.
Author Response
Responses to reviewer 1
The Authors improved the manuscript, although I still have some remarks.
I still don't understand the purpose of 'box 1' - couldn't it be another numbered paragraph?
-Response: Thank you for your kind suggestion. We consider the content of box 1 as supplementary knowledge for the main text which is specifically focused on roles of psychological factors in physiological condition. Thus, we think it might be more understandable and organized to use a box to show comparing to put it directly in the main text.
I still don't see the materials and methods section. The Authors say that they added such part, but I can't really find this in the text. Please add a distinct section and describe how you searched articles, what MeSH you used and what languages were taken into account.
- Response: Thank you for your kind advice. We did not include detailed search strategy for it was not designed to be a systematic review or meta-analysis from its beginning. But we have added the mind map to show the outline of our review(Please refer to the attached file).
References are still not prepared according to Journal's guidelines. In the text they are also superscripted, not in the square brackets.
Response: Thank you for your kind remind. We have modified the all reference formats according to the Journal's guidelines.Thanks again for your generous comments.